# Folate-Functionalization Enhances Cytotoxicity of Multivalent DNA Nanocages on Triple-Negative Breast Cancer Cells

**DOI:** 10.3390/pharmaceutics14122610

**Published:** 2022-11-26

**Authors:** Valeria Unida, Giulia Vindigni, Sofia Raniolo, Carmine Stolfi, Alessandro Desideri, Silvia Biocca

**Affiliations:** 1Department of Systems Medicine, University of Rome Tor Vergata, Via Montpellier 1, 00133 Rome, Italy; 2Department of Biology, University of Rome Tor Vergata, Via della Ricerca Scientifica 1, 00133 Rome, Italy

**Keywords:** DNA nanostructure, AS1411, miR-21, folate receptor, nucleolin, doxorubicin delivery

## Abstract

DNA is an excellent programmable polymer for the generation of self-assembled multivalent nanostructures useful for biomedical applications. Herein, we developed (i) folate-functionalized nanocages (Fol-NC), very efficiently internalized by tumor cells overexpressing the α isoform of the folate receptor; (ii) AS1411-linked nanocages (Apt-NC), internalized through nucleolin, a protein overexpressed in the cell surface of many types of cancers; and (iii) nanostructures that harbor both folate and AS1411 aptamer functionalization (Fol-Apt-NC). We analyzed the specific miRNA silencing activity of all types of nanostructures harboring miRNA sequestering sequences complementary to miR-21 and the cytotoxic effect when loaded with doxorubicin in a drug-resistant triple-negative breast cancer cell line. We demonstrate that the presence of folate as a targeting ligand increases the efficiency in miR-21 silencing compared to nanocages functionalized with AS1411. Double-functionalized nanocages (Fol-Apt-NC), loaded with doxorubicin, resulted in an increase of over 51% of the cytotoxic effect on MDA-MB-231 cells compared to free doxorubicin, demonstrating, besides selectivity, the ability of nanocages to overcome Dox chemoresistance. The higher efficiency of the folate-functionalized nanocages is due to the way of entrance, which induces more than four times higher intracellular stability and indicates that the folate-mediated route of cell entry is more efficient than the nucleolin-mediated one when both folate and AS1411 modifications are present.

## 1. Introduction

A key research effort in nanomedicine is the development of universal drug carriers that can harbor multiple ligands for selective targeting, diagnosis, biosensing, and therapy. In the last two decades, the field of DNA nanotechnology has undergone a tremendous expansion and produced a variety of wireframes and single-layered and solid nanostructures. DNA nanostructures display important advantages when compared to other nanomaterials, such as biocompatibility, stability, and versatility. Concerning biomedical applications, DNA nanotechnology offers easy design techniques for the sequence-specific self-assembly of nucleic acids into complex structures and has high potential in enhancing selective drug targeting and reducing drug toxicity. Stability can be modulated depending on the assembly of covalent or noncovalent nanostructures, DNA strands can be functionalized with cellular recognition signals for selective cell or tissue targeting, size and geometry can be highly modulated, and nanostructures can be engineered with multifunctional motifs [1,2,3,4]. In the last years, we have been involved in studying the design, assembly, and structural–dynamical properties of DNA-based truncated octahedral nanocages, identifying the parameters that can modulate the yield of assembly and morphology [5,6,7] for their use as anti-cancer agents. We characterized different types of fully covalent and multivalent DNA nanocages and studied the in vitro and in-cell stability, their receptor-mediated cell entry, and their activity as selective drug carriers [8,9,10,11,12]. By introducing four miRNA sequestering units in a corner of a truncated octahedral DNA nanocage, we have obtained an openable nanostructure able to bind specific miRNA and function as sponges for selective oncomiR sequestering, such as miR-21 [12,13]. Thus, the up-regulation of miR-21 was consistently found in clinical samples from cancer patients. In cancer cells, the increase in miR-21 expression levels was found to regulate different apoptotic genes and to down-regulate PTEN and programmed cell death protein 4 (Pdcd4), which are tumor suppressor proteins [14,15]. DNA nanocages, harboring sequences complementary to miR-21, sequester miR-21 within cells, leading to a reduction of over 80% in different cancer cells after 48 h treatment [13].

When functionalized with targeting signals, such as folate or the G-rich aptamer AS1411, DNA nanocages do not alter their morphology and are selectively targeted to tumor cells [9,10,16]. We focused on the α-isoform of the folate receptor (αFR) since it is a tumor-associated antigen overexpressed in many tumors and largely absent in normal tissues. The conjugation of folate with DNA nanocages does not alter the high affinity for its receptor, leading to high targeting efficiency [9,13]. Likewise, the AS1411 aptamer is largely used as a targeting signal in nanotechnology [17,18]. The introduction of the AS1411 aptamer allows the nanocage to selectively bind to nucleolin, a multifunctional cyto-nucleoplasmic protein mostly located in the nucleoli in normal cells and delocalized on the cell surface and aberrantly overexpressed in many types of cancers [19,20]. Notably, AS1411 also has well-characterized toxic cell effects. It belongs to a unique class of aptamers constituted of G-rich sequences that, under physiological conditions, spontaneously fold into non-canonical four-stranded structures named G-quadruplexes. The interaction of the G-rich AS1411 with nucleolin leads to the inhibition of proliferation and the cell death of cancer cells, with little effect on normal cells [18,21]. These effects are also related to the role of nucleolin in promoting the maturation of a specific set of oncomiR, which includes miR-21, implicated in the pathogenesis of several human cancers [22]. For its properties as an inhibitor of nucleolin, the AS1411 aptamer has been used for its anticancer therapeutic effect in several clinical trials [18], and is linked to nano-formulations for targeted drug delivery [23,24,25,26,27,28]. Recently, we have reported that octahedral nanocages functionalized with one AS1411 aptamer molecule (Apt-NC) are selectively internalized by nucleolin-positive cancer cells and show an over 200-fold increase in anti-cancer activity when compared to treatment with the free aptamer [16].

Doxorubicin (Dox) is a frontline chemotherapeutic agent used to treat many hematological malignancies and a wide variety of solid tumors [29]. Dox intercalates between DNA bases and, when inside cells, induces many toxic effects including the disruption of topoisomerase-II-mediated DNA repair and the generation of reactive oxygen species, which result in the inhibition of DNA replication and cell death [30]. To avoid debilitating effects due to high dose administration and systemic toxicity, due to the lack of cancer cell selectivity of free Dox, we have successfully loaded folate-functionalized DNA nanocages with Dox, exploiting the DNA-intercalating properties of this molecule. Dox-loaded folate functionalized octahedral nanocages selectively deliver Dox to the folate receptor α-enriched cancer cells [9,13].

Breast cancer often presents the expression of three markers that allow the classification of these tumors, which are the estrogen receptor (ER), progesterone receptor (PR), and amplification of HER-2/Neu. Triple-negative breast cancer (TNBC) are tumors that lack all these three molecular markers. These tumors are known to have a poor prognosis due to the lack of targeted therapies and higher rates of treatment resistance [31]. The MDA-MB-231 cell line is an epithelial, human breast cancer cell line, which represents a good model of triple-negative breast cancer and of inducible doxorubicin chemo-resistance. As reported for other cancers, the oncomiR miR-21 is overexpressed in TNBC [32,33].

Here, we compare multifunctional nanostructures for drug delivery, namely folate-functionalized nanocages (Fol-NC), AS1411-linked nanocages (Apt-NC), and nanostructures that harbor both folate and AS1411 (Fol-Apt-NC) in MDA-MB-231 cells. The presence of folate improves the miR-21 silencing activity of nanocages harboring the anti-miR sequences and the cytotoxic effect of Dox-loaded Fol-Apt-NCs. Fol-Apt-NCs are also more efficient in killing cancer cells than free doxorubicin, demonstrating that the polyvalent nanocages are able to overcome Dox chemoresistance of the triple-negative breast cancer cell line MDA-MB-231.

## 2. Materials and Methods

### 2.1. Preparation of Functionalized DNA Nanocages

Nanocages (NC), folate-functionalized (Fol-NC), AS1411 aptamer-functionalized (Apt-NC), folate and aptamer-modified (Fol-Apt-NC), and nanocages assembled with sequences complementary to mature miR-21-5p (NC-*anti*miR21, Fol-NC-*anti*miR21; Apt-NC-*anti*miR21 and Fol-Apt-NC-*anti*miR21) were prepared as described [12]. Briefly, nanocages were assembled by mixing equimolar amounts of 8 oligonucleotides, reported in Appendix A, in TAM buffer (40 mM Tris–acetic acid, pH 7.0, 12.6 mM magnesium acetate) and then incubated for 2 h at 25 °C with T4 DNA ligase (New England Biolabs Inc., Ipswich, MA, USA) to covalently link the obtained octahedral structures. The correctly assembled nanostructures were purified by polyacrylamide gel separation, eluted, and 2-propanol precipitated.

The sequences of the oligonucleotides used for the assembly of all described NC are reported in Appendix A. The oligonucleotide 6 (OL6) was modified by adding a biotin molecule to allow the detection of nanostructures through the streptavidin–biotin reaction.

### 2.2. Cell Cultures

MDA-MB-231 cells, derived from human breast adenocarcinoma, were grown in RPMI 1640 (Euroclone, Devon, UK) supplemented with 10% FBS (Gibco, Paisleg, UK), 1 mM L-glutamine (Sigma Aldrich, St Louis, MO, USA), and 100 U/mL penicillin–streptomycin (Euroclone, Devon, UK). For DNA nanocage experiments, cell culture medium was replaced with folate-free RPMI 1640 supplemented with 5% FBS.

### 2.3. DNA Nanocages Stability and DNA Blot

For in vitro stability, biotinylated nanocages were incubated in culture medium supplemented with 10% FBS at 37 °C for different times. Each sample was then digested with proteinase K (100 µg/mL) for 1 h at 37 °C and analyzed by DNA blot, as described [8].

For evaluating the intracellular stability, cells were plated in 48-well plates at a density of 3 × 10^4^ cells/well and grown for 24 h before treatments. After incubation with DNA nanocages for different time periods, cells were lysed, centrifuged, digested with proteinase K, and analyzed with DNA blot [8]. Biotinylated nanocages detection was carried out using streptavidin–HRP (horseradish peroxidase) (Abcam Inc., Toronto, ON, Canada), and visualized using enhanced chemiluminescence (ECL Extend, Euroclone, Devon, UK). For image processing, photographic films were digitized and densitometric analysis was performed with ImageJ 1.52a software.

### 2.4. RNA Isolation and qPCR for miRNA Expression Analysis

For RNA extraction, MDA-MB-231 was plated in 48-wells plate at a density of 3 × 10^4^ cells/well and grown 24 h before treatments. After treatment with nanocages for different time points, total RNA was extracted using RNeasy Mini Kit (Qiagen, Hilden, Germany) and quantified using NanoDrop spectrophotometer (NanoDrop ND-1000, Waltham, MA, USA).

RNA was reverse-transcribed into cDNA by using miRCURY LNA RT Kit (Qiagen, Hilden, Germany). For quantitative analysis of miR-21 expression, qPCR amplification of cDNA was performed using miRCURY LNA miRNA PCR Assay (Qiagen, Hilden, Germany) on a Real-Time PCR Detection System (Bio-Rad, Hercules, California, CA, USA), following the manufacturer’s instruction. Relative transcript quantification of miR-21 was determined using the ΔΔCt method, normalized to the levels of endogenous U6 snRNA and to the untreated control cells.

### 2.5. DNA Nanocages Intercalation with Doxorubicin

Doxorubicin (Cell Signaling Technologies) was diluted in TBS (Tris-HCl 50 mM, NaCl 150 mM, pH 7.4) at a concentration of 200 μM. DNA nanocages were incubated overnight at room temperature (RT) with Dox at 1:2 Dox:base-pairs (bp) ratio [9]. The Dox-loaded nanocages were purified from the unloaded Dox by gel filtration through Sephadex G-25, using illustra MicroSpin G-25 columns (GE Healthcare, Chicago, IL, USA). Fluorescent intensity was determined by measuring Dox fluorescence using a Spark^®^ Multimode Microplate Reader (TECAN, Männedorf, Switzerland) (Excitation: 485 nm; Emission: 590 nm). The total amount of intercalated Dox in nanocages was calculated from the fluorescence intensity of the sample after acidification at pH 2, which causes total Dox release from DNA, and compared with a calibration curve shown in Appendix A, obtained from measurements on free Dox samples at known concentration [13]. The percentage of Dox intercalated in nanocages was calculated using the following formula:(1)$$\frac{{\mathrm{Dox}}_{\mathrm{released}}}{{\mathrm{Dox}}_{\mathrm{Tot}}}\times 100$$
where Dox_released_ is the concentration of intercalated Dox after sample acidification at pH 2 and Dox_Tot_ is the concentration of Dox in the initial intercalation mix. The total amount of intercalated Dox in all functionalized DNA nanocages after gel purification is similar and is 22.2 ± 1.2% of the Dox initial concentration.

### 2.6. Cell Viability Assay

MDA-MB-231 were plated in 96-well plates at a density of 15 × 10^3^ cells/well. After 24 h, cells were treated with unloaded or Dox-loaded nanocages for different times at 37 °C. Time and dose-dependent toxicity of free Dox were assessed on MDA-MB-231 and HeLa cells (Appendix A). The MTS assay was performed using the CellTiter 96 Aqueous One Solution Cell Proliferation Assay (Promega, Madison, WI, USA). Spark microplate reader (Tecan Trading AG, Männedorf, Switzerland) was used for measuring the absorbance at 492 nm. Cell viability of treated cells was normalized to the control condition (untreated cells).

### 2.7. Statistical Analysis

All experiments were carried out in triplicate. Results were expressed as mean ± S.E.M and statistical analyses were performed with ANOVA test in GraphPad Prism. Differences were considered statistically significant when *p* < 0.05 (*), *p* < 0.01 (**), and *p* < 0.001 (***).

## 3. Results

### 3.1. Design and In Vitro Stability of Nanocages

For receptor-mediated cell uptake, one of the eight oligonucleotides used for the assembly of the nanocages was functionalized with folate, as described in [9,10], or with the aptamer AS1411, as described [16], to obtain folate- or aptamer-functionalized nanocages. A double-functionalized nanocage, which harbors folate and AS1411 aptamer modification (Fol-Apt-NC), was assembled using both the folate-functionalized oligonucleotide and the AS1411-linked oligonucleotide (Appendix A). For miR-21 silencing, *anti*miR21-nanocages were assembled with four DNA hairpin units, complementary to mature miR-21-5p, introduced in one truncated face of the octahedral DNA structure [13]. The binding of specific miRNA to the structure leads to a conformational change of the nanocage and to the consequent increase in the molecular weight, detectable by gel electrophoresis, as shown in Appendix A. A 21-nucleotide scramble DNA sequence was assembled in place of the anti-miR-21 hairpin to build the nanocages used as negative controls.

Figure 1A shows the schematic representation of the four nanostructures used in this work, which harbor four DNA hairpins containing a 21-nucleotide scramble DNA sequence or a complementary miR-21 sequence for miR-21 silencing. All nanocages were also functionalized with a biotin molecule (Bio) for their detection through the biotin–streptavidin assays. Due to the high affinity of streptavidin for biotin, the system has a sensitivity up to 100 times higher than the ethidium bromide staining, allowing the analysis of their size, morphology, and in vitro stability in the presence of serum. For the analysis of their in vitro stability, all nanocages were incubated in 10% fetal bovine serum (FBS) for 1, 5, 24, and 48 h at 37 °C and analyzed with a DNA blot, shown in Figure 1B. At time zero, the four nanostructures have identical mobility indicating that they are intact and have an identical size and morphology. All nanostructures have a similar half-life of 22.3 h, calculated by the relative intensity of each band visualized through a DNA blot, which is comparable to the previously reported half-life of pristine DNA nanocages (NC) and Fol-NCs [13].

### 3.2. Selective miR-21 Silencing by Functionalized-Nanocages

Fol-NC, Apt-NC, and Fol-Apt-NC were assembled with scramble hairpins and Fol-NC-*anti*miR21, Apt-NC-*anti*miR21, and Fol-Apt-NC-*anti*miR21 with hairpins complementary to miR21 using OL_scr_ and OL_miR21_, respectively, shown in Appendix A. For the evaluation of selective miR-21 sequestering activity, qPCR analysis of miRNA expression level was carried out using the ΔΔCt method, normalized to the levels of endogenous U6 snRNA. Figure 2 shows the reduction of miR-21 expression in MDA-MB-231 cells induced by 15 nM of differently modified nanocages, incubated for 24 h and 48 h. It is worth noting that 15 nM is a concentration that leads to efficient miR-21 silencing and, as shown in Appendix A, very low cell toxicity in the time period of the experiment. Fol-NC-*anti*miR21 incubation leads to 62.5 ± 3% reduction of miR-21 expression at 24 h and 75 ± 10% at 48 h, demonstrating the high efficiency of the folate functionalized nanocages and confirming previous results obtained in HeLa cells [13]. Cells incubated with nanocages functionalized with the AS1411 aptamer (Apt-NC) display a lower miR-21 reduction, being 47.5 ± 8% at 24 h, which is even lower (40 ± 16%) after 48 h. Cells incubated with nanocages functionalized with both folate and AS1411 (Fol-Apt-NC) display a miR-21 reduction of 35 ± 6% and 56.7 ± 3% at 24 and 48 h incubation, respectively.

It is interesting to note that the most striking effect is observed with the folate functionalized nanocages, despite the fact that free AS1411 aptamer, being a nucleolin inhibitor, downregulates several miRNAs, including miR-21, in cancer cells [22]. Accordingly, miR-21 expression in MDA-MB-231 cells treated with 10 µM free AS1411 is reduced by 37.5% compared to non-treated control cells (Figure 2B). In line, Fol-Apt-NC, i.e., nanocages without the antimiR21 sequestering units, have an effect on the level of miR-21, indicating that AS1411 displays its antimiR-21 activity in folate-functionalized nanocages. The anti-miR21 effect is not observed in cells incubated with Apt-NC, indicating that the miR-21 reduction due to the aptamer functionalization on the cage is observable only in the presence of folate.

To verify the downstream effect of miR-21 knockdown in MDA-MB-231 cells, we analyzed the expression level of PTEN, a gene regulated by miR-21 level by Western blot (Appendix A). Densitometric analysis shows that PTEN level is about 2-fold and 1.5-fold higher, respectively, in Fol-NC-*anti*miR21 and Apt-NC-*anti*miR21 treated cells vs. negative controls, confirming the crucial role of folate in the miR-21 reduction and in the consequent effects.

### 3.3. Cytotoxic Effects of Fol-NC, Apt-NC, and Fol-Apt-NC, Loaded or Not with Doxorubicin

For the analysis of the cytotoxic effect, nanocages were assembled without anti-miR21 sequestering units. MDA-MB-231 cells were treated with Fol-NC, Apt-NC, and Fol-Apt-NC, loaded or not with Doxorubicin, and their effect on cell viability was investigated in comparison with the effect of pristine nanocages and free Dox at different time periods (Figure 3). The concentration of nanostructures used was 11 μM (base pair concentration), corresponding to 45 nM, which leads to the 9 ± 2% and 13.5 ± 0.5% reduction of cell viability at 24 h incubation for Apt-NC and Fol-Apt-NC, respectively, as it can be seen from the dose-dependent curve of cell viability in the presence of nanocages shown in Appendix A. Cells were also treated with free Dox at a concentration of 1.2 µM, which is the Dox concentration loaded into DNA nanocages, calculated as described in Materials and Methods and in Appendix A. Figure 3 shows that all types of nanocages, including pristine nanocages (NC), behave as good Dox carriers with similar antiproliferative activity in MDA-MB-231 cells, but the strongest cytotoxic effect is observed treating cells with Dox-loaded Fol-Apt-NC. In this case, a synergic effect, due to the double presence of folate and AS1411 aptamer, is observed, reaching a value of 47 ± 7% and 74.5 ± 3.5% reduction in cell viability after 48 and 72 h incubation, respectively. Conversely, MDA-MB-231 cells treated with Dox-loaded Apt-NC display a reduction in cell viability of 38 ± 4% and 50 ± 1% after 48 and 72 h, respectively. The comparison between Dox-loaded Fol-Apt-NC and Dox-loaded Apt-NC, which only differ in the presence of a folate molecule, highlights an increase of 50 ± 1% in the cytotoxicity after treatment with Fol-Apt-NC for 72 h.

Notably, treatment with Dox-loaded Fol-Apt-NC nanocages leads to a 45% increase in cytotoxicity when compared to free Dox in MDA-MB-231 cells, demonstrating that double-functionalized NCs are able to overcome Dox-resistance of the breast carcinoma cell line. The time and dose-dependent curves of toxicity of free Dox in MDA-MB-231 are shown in Appendix A, compared to the curves obtained with the Dox-sensitive HeLa cell line.

To investigate the mechanism of Dox delivery by nanocages we compared the intracellular localization of Dox in MDA-MB-231 incubated with free Dox or Dox-loaded Fol-Apt-NC, exploiting the natural red fluorescence emission of Dox. The confocal analysis is shown in Appendix A. While free Dox is clearly localized in the nuclei (Appendix A, left panel), when Dox is delivered through Fol-Apt-NC, it also localizes in the cytoplasm (Appendix A, right panel), confirming a receptor-mediated entry mechanism and different intracellular traffic of Dox.

Figure 3 also shows the effect of nanocages not-loaded with Dox on MDA-MB-231. Interestingly, the treatment of cells with Fol-Apt-NC induces the highest cytotoxic effect compared to the other NCs, already detectable after 24 h incubation (13.5 ± 1.7%) which reaches a value of 31 ± 2% and 52 ± 6% reduction of cell viability at 48 and 72 h incubation, respectively. As expected, incubation with pristine (NC) or Fol-NC was not cytotoxic to MDA-MB-231 cells, as previously reported in other cell lines [9], while a weak but statistically significant reduction of cell viability of 19.7 ± 5.7% was detected with Apt-NC at 72 h treatment, confirming in MDA-MB-231 the cytotoxic effect of AS1411-linked to nanocages, as reported in HeLa cells [16].

### 3.4. Intracellular Stability of Functionalized DNA Nanocages

Considering that the folate is a non-toxic, very efficient targeting molecule that does not induce cytotoxicity even when Fol-NCs are used up to 12 μg/mL (60 nM) concentration [9], one of the possible explanations for the higher cytotoxic effect of Fol-Apt-functionalized nanocages (Figure 3) is that the nanocages with double-functionalization (folate and AS1411) enter preferably via the folate route, rather than via nucleolin, and are, consequently, more stable inside cells.

To verify this hypothesis, we compared the intracellular stability in MDA-MB-231 cells of non-functionalized nanocages (NC), folate-functionalized (Fol-NC), AS1411-linked nanocages (Apt-NC), and AS1411 and folate doubly functionalized nanocages (Fol-Apt-NC). In detail, cells were treated with 8 μg/mL (45 nM) of biotinylated nanocages and, at different time points, nanocages were purified from cell extracts and analyzed with a DNA blot using streptavidin–HRP (Figure 4). It is important to remark that we did not observe any difference in the internalization mechanism using a range of concentrations between 1.5 and 10 μg/mL nanostructures. Lanes 1 and 5 of panel A and 1 and 5 of panel B in Figure 4 show the electrophoretic mobility of the input band of NC and Fol-NC (15 ng), respectively, prior to the incubation with cells (time 0). In MDA-MB-231 cells, purified DNA nanocages run as a single band with mobility comparable to the input, indicating that they are intact inside cells. It is worth noting that, in MDA-MB-231 cells, bands corresponding to pristine DNA nanocages (NC) are only barely detectable, indicating a very low amount of NC inside cells (Figure 4, panel A, lanes 1-4). On the contrary, after 3 h incubation with cells, a band corresponding to intact Fol-NC (Figure 4, panel A, lanes 6), or Apt-NC (panel B, lanes 2) and Fol-Apt-NC (panel B, lane 6), is detectable. A band with similar intensity is visible at 24 h of incubation (Figure 4, panel A, lane 7; panel B, lanes 3 and 7). At 48 h incubation, only cells treated with Fol-NC or Fol-Apt-NC (Figure 4, panels A and B, lanes 8) show an intense band indicating a marked amount of intact nanocages inside MDA-MB-231 cells. In the case of the incubation of cells with Apt-NC at 48 h, the band is almost undetectable, confirming our previous observation that AS1411 functionalized nanocages, when uptaken by the nucleolin-mediated pathway, are degraded and do not accumulate inside cells [16].

The amount of internalized Fol-Apt-NC at 48 h, evaluated by densitometric analysis (shown in lower panels of Figure 4), is 98 ± 10 ng/10^6^ cells, which is more than four times the amount of internalized Apt-NC (22 ± 4 ng/10^6^ cells). This result not only confirms the higher intracellular stability of nanocages functionalized with folate but also that the folate-mediated route of cell entry is more efficient than the nucleolin-mediated one when both folate and AS1411 modifications are present.

## 4. Discussion

Here, we have compared the efficacy of multifunctional octahedral DNA nanocages harboring two ligands for selective targeting, namely folate and AS1411 aptamer, and engineered with four sequestering sequences complementary to miR-21 for inhibiting miR-21 function. qPCR assays for the analysis of miR-21 expression level showed that the presence of folate confers higher efficacy on the reduction of miR-21 level, comparable to what was observed in other cancer cell lines [13]. In detail, folate-functionalized nanocages lead to a 75% reduction of miR-21 level compared to a 50% reduction of nanocages functionalized with the G-rich aptamer AS1411 in the first 2 days of treatment. It is worth noting that the presence of both folate and aptamer AS1411 in the nanocages (Fol-Apt-NC-*anti*miR21) does not improve the efficacy of miR-21 down-regulation as compared to Apt-NC-*anti*miR21 at 24 h, although the effect is more evident for a longer time when folate is present (compare the effect at 24 vs. 48 h between Apt-NC-*anti*miR21 and Fol-Apt-NC-*anti*miR21 in Figure 2). This result is surprising considering that free AS1411, selectively inhibiting the intracellular activity of nucleolin, has an inhibitory effect on the expression of various miRNAs, including miR-21 [22]. Nucleolin promotes the maturation of a specific set of miRNAs that are overexpressed in several human cancers, such as miR-21, miR-103, miR-221, and miR-222. Their high levels are often associated with greater aggressiveness of tumors and resistance to antineoplastic therapies [34,35,36,37]. One might expect that nanocages that display both AS1411 aptamer and antimiR21 sequestering units (Apt-NC-*anti*miR21) would have an additive inhibitory effect on miR-21 level, which, instead, is not observed. A possible explanation is that AS1411 has a prevalent cytotoxic effect in MDA-MB-231 cells and, therefore, it activates apoptotic pathways rather than miRNA inhibition. Another hypothesis stems from the mechanism of the action of AS1411 and the multi-functional and dynamic nature of nucleolin. AS1411 binding to nucleolin affects the molecular interactions between nucleolin and its many binding partners [21]. In particular, the binding of AS1411 to nucleolin reduces the level of mature miR-21 [22], which is the target for NC’s sequestering units. These two different mechanisms could coexist in nanocages harboring anti-miR21 sequences and result in a paradoxical feedback effect, where the efficacy of miR-21 sequestering seems to lower over time.

The most striking positive effect due to the presence of folate is observed by comparing the doxorubicin delivery and its anti-tumor activity when loaded into the differently functionalized nanocages. Treatment of the high-invasive Dox-resistant breast carcinoma MDA-MB-231 cells with Fol-Apt-NC loaded with Dox leads to an increase between 45 and 50% in the cytotoxic effect over all other nanostructures. It is interesting that, in these cells, Dox-loaded Fol-Apt-NC nanocages also have higher cytotoxicity (over 45%) when compared to free Dox, demonstrating that treatment with double-functionalized NC overcomes the Dox-resistance of MDA-MB-231. Confocal analysis indicates a cytoplasmic localization of Dox in MDA-MB-231 when delivered through Fol-Apt-NC (Appendix A), as previously shown for Fol-NC in HeLa cells [9], which suggests that the receptor-mediated entry enables Dox to escape the recognition by P-gp proteins on the cell membrane enhancing Dox accumulation in resistant cancer cells without efflux [9,38].

One consideration to explain the advantage of using Fol-Apt-NC is that they enter preferably via the folate receptor α route, rather than via nucleolin, and are more stable inside cells. We have demonstrated that when folate is used as a targeting ligand and linked to DNA nanostructures, it permits the nanocages to enter through the αFR, conferring stability for hours and letting them slowly accumulate inside cells without degradation [10,13], exploiting the fact that folate is internalized in the cytoplasm for cellular utilization and not for degradation [39]. The comparison of the intracellular stability of differently functionalized nanocages in MDA-MB-231 clearly indicates the high and comparable intracellular persistence of Fol-NC and Fol-Apt-NC (Figure 4). The highest number of intact Fol-NCs and Fol-Apt-NCs inside cells occurs after 48 h, whilst Apt-NCs reach their highest intracellular concentration after 3 h incubation and then they slowly decrease. On the contrary, the in vitro stability of Apt-NC and Fol-Apt-NC in serum is very similar over time (Figure 1B).

The aptamer AS1411 confers a selective targeting to Apt-NC, compared to pristine nanocages, and its presence in Apt-NC is responsible for the high efficiency of entry to nucleolin-positive cancer cells [16]. On the other hand, the presence of AS1411 aptamer appears to interfere with the intracellular stability of the nanocages because the nucleolin-mediated entry, using a flotillin-dependent endocytosis mechanism, traffic through the endo-lysosomal degradation pathway [16]. Notably, after 48 h, Fol-Apt-NCs accumulate over four times more than Apt-NCs in MDA-MB-231 cells (98 ± 10 ng Fol-Apt-NC/10^6^ cells vs. 22 ± 4 ng Apt-NC/10^6^ cells) demonstrating that, although they harbor two targeting signals, they preferably enter through the folate receptor, accumulate into the cells, and carry out the cytotoxic activity.

In addition to stability, another crucial aspect to consider for explaining the higher cytotoxic effect is the possible activation of different pathways depending on the mechanism of entry into the cells. The transcriptomic analysis of cells treated with the different types of nanocages is under investigation to highlight whether the nanocages that entered through different receptor-mediated mechanisms also activate different signaling and toxic pathways.

In conclusion, the high versatility of DNA, which allows the possibility of constructing nanostructures with multiple targeting ligands and active motifs, broadens the possibilities of choice regarding the therapeutic targets and highlights the importance of the presence of modifications, such as folate, for selective miRNA sequestering and prolonged release of anti-tumor drugs.

## Figures and Tables

**Figure 1 pharmaceutics-14-02610-f001:**
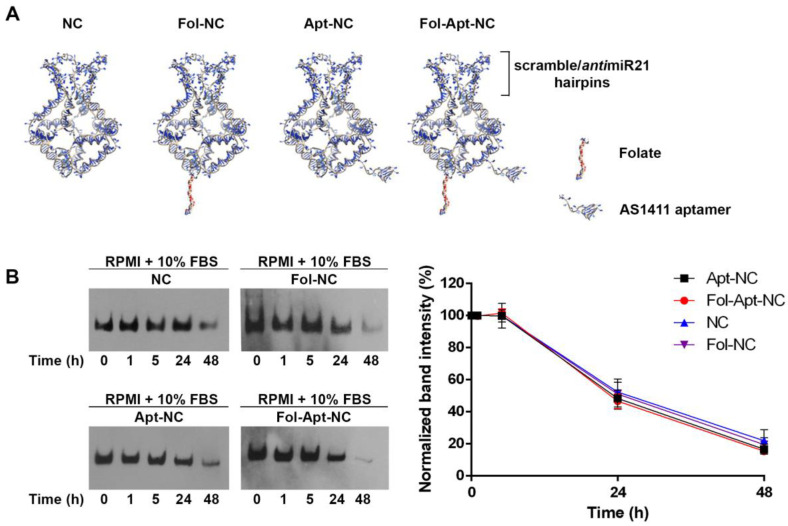
(**A**) Schematic representation of octahedral nanocages (NC), folate-functionalized nanocages (Fol-NC), AS1411-linked nanocages (Apt-NC), and both folate and AS1411 functionalized nanocages (Fol-Apt-NC), harboring the scramble or anti-miR21 hairpins. (**B**) DNA blot analysis. Biotinylated nanocages are detected using Streptavidin–HRP. 30 ng of nanocages before incubation with serum proteins (time 0) are shown in each panel. The densitometric analysis (shown in the right panel) was performed using ImageJ software. The relative intensity of each band was normalized to the intensity of the band corresponding to each time 0 and reported in the graph. The graph shows average ± SEM of three different experiments.

**Figure 2 pharmaceutics-14-02610-f002:**
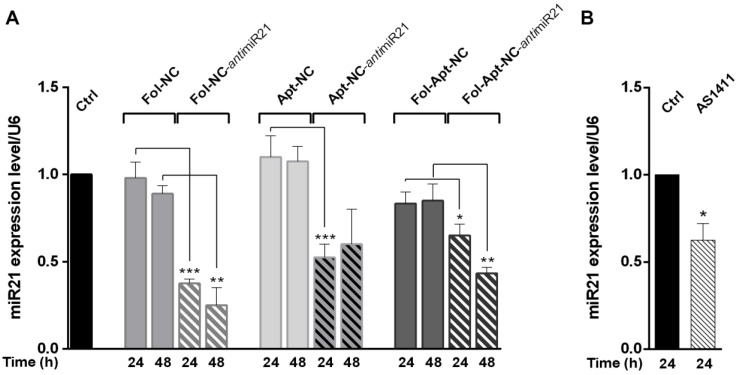
MiR-21 silencing activity of folate and AS1411-functionalized nanocages in MDA-MB-231 cells. (**A**) qPCR analysis of miR-21 expression level after treatments of MDA-MB-231 cells for 24 and 48 h, with different DNA nanocages carrying sequences complementary to miR-21 (antimiR21) or with non-relevant DNA sequences as negative control. (**B**) miR-21 expression level after treatment of MDA-MB-231 cells with 10 μM free AS1411. Values are expressed as mean ± S.E.M. of three different independent experiments. Statistical significance: * *p* < 0.05, ** *p* < 0.01, and *** *p* < 0.001 (ANOVA test).

**Figure 3 pharmaceutics-14-02610-f003:**
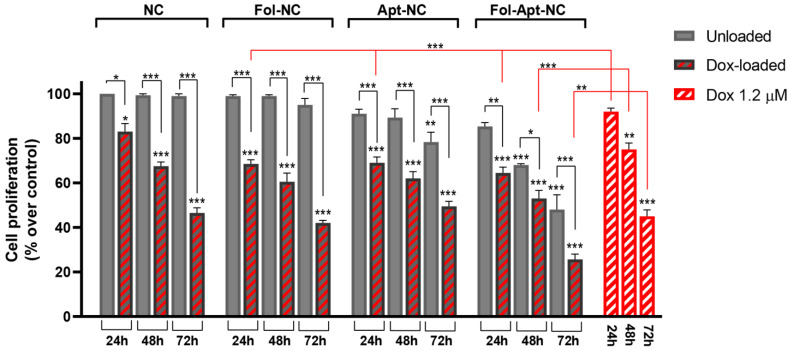
Cytotoxic effect of Dox-loaded or not-loaded NC, Fol-NC, Apt-N,C and Fol-Apt-NC and of free Dox in MDA-MC-231 cells. Cell proliferation of cells treated with NC, Fol-NC, Apt-NC, and Fol-Apt-NC not loaded (light gray) or loaded with Dox (gray and red striped) and free Doxorubicin (red and white striped) at different times, as indicated, was evaluated with MTS assay. The data represent mean ± S.E.M. of three separate experiments. The values are the mean of six replicates, normalized on cell proliferation of untreated cells. Statistical analysis was performed to compare at each time point (i) all groups vs. untreated cells, (ii) Dox-loaded-NC groups vs. unloaded-NC treated cells, and (iii) free Dox treated vs. Dox-loaded-NC treated cells. Statistical significance: * *p* < 0.05, ** *p* < 0.01, and *** *p* < 0.01 (ANOVA test).

**Figure 4 pharmaceutics-14-02610-f004:**
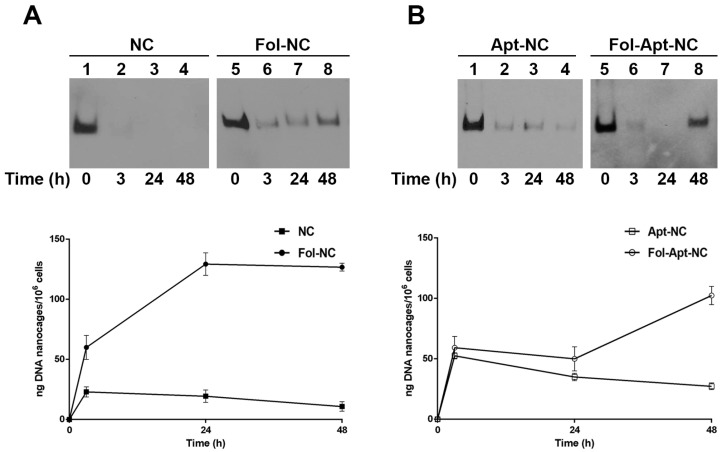
Intracellular stability of functionalized nanocages in MDA-MB-231 cells. Representative DNA blots of MDA-MB-231 cell extracts incubated for 3, 24, and 48 h with (**A**) NC (lanes 2, 3, and 4) or Fol-NC (lanes 6, 7, and 8) and (**B**) Apt-NC (lanes 2-3 and 4) and Fol-Apt-NC (lanes 6, 7 and 8). Biotinylated nanostructures were detected with streptavidin (HRP)–biotin reaction. 15 ng of DNA nanocages before incubation with cells (time 0) are shown in lanes 1 and 5. Histograms in lower panels show the amount of DNA nanocages, expressed in ng/10^6^ cells, internalized in cells at different time points at 37 °C, as indicated. For image processing and densitometric analysis, bands were analyzed by using ImageJ software. Values, calculated by using GraphPad Prism, were expressed as a mean ± SEM.

## Data Availability

Not applicable.

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
