# Peer review of "Folate-Functionalization Enhances Cytotoxicity of Multivalent DNA Nanocages on Triple-Negative Breast Cancer Cells"

_pharmaceutics, 2022, doi:10.3390/pharmaceutics14122610_

Round 1

Reviewer 1 Report

The authors reported three functionalized nanocages(Fol-NC, Apt-NC, Fol-Apt-NC) and analyzed the silencing activity of all types of nanostructures of the miRNA and the antitumor effect of Fol-Apt-NC when loaded with doxorubicin on the chemo-resistant triple negative breast cancer cell line. In addition, the authors studied the ability of nanocages to overcome Dox-chemoresistance. However, there are several problems listed as follows:

Major points:

1. Figure 1A shows the schematic representation of the four nanostructures, carrying the miRNA sequestering units.There is no relevant characterization data on whether the size and morphology of nanocages modified by different methods are changed, and whether the changes will affect the biological activity?

2. Why the dosage of Fol-Apt-NC was different on the selective miR-21 silencing (15 nM) and cytotoxic effect (45 nM)?  There should be more experiments and discussion

3. Fol-Apt-NC loaded with Dox exhibited good efficacy on cell model, if possible, the in vivo experiments are suggested to explore.

4. In this manuscript, there were a few studies about overcoming Dox chemoresistance of Fol-Apt-NC loaded with Dox, and more experiments should be investigated to further verify the mechanism.

5. The method of this manuscript is far from the demand to repeated the experiments. More details are needed.

Reviewer 2 Report

The manuscript entitled as “Folate-functionalization enhances cytotoxicity of multivalent DNA nanocages on triple negative breast cancer cells” written by Unida et al. presents an original idea and great work for the cancer research. However, I found cytotoxicity results weak, compared to the other results.

Here I suggest several points for consideration.

1) Dox loading calculation is not clearly explained. Authors should explain how they decided to use 1.2 µM of Dox.

2) All blots (n=3) should be added as supplementary images.

3) In the introduction giving more details about miR21 would be good. Similarly, AS1411 and nucleolin mediated entry etc. in the abstract.

4) It is not correct to write "anti-tumor effects", because the results only show in vitro anti-proliferative effects.

5) By looking  at the Figure S3, the relation between PTEN and apoptosis in the presence of dox loaded nano-cages could be also shown.

6) In figure 3, it is important to compare the significance between different treatment groups not only with the untreated control group.

7) In Figure 4, legends should be added for the graphs, it is not clear.
